# Beta-Barrel Channel Response to High Electric Fields: Functional Gating or Reversible Denaturation?

**DOI:** 10.3390/ijms242316655

**Published:** 2023-11-23

**Authors:** Ekaterina M. Nestorovich, Sergey M. Bezrukov

**Affiliations:** 1Department of Biology, The Catholic University of America, Washington, DC 20064, USA; 2Section on Molecular Transport, *Eunice Kennedy Shriver* National Institute of Child Health and Human Development, National Institutes of Health, Bethesda, MD 20892, USA; bezrukos@mail.nih.gov

**Keywords:** protein folding, single-molecule experiments, channel reconstitution, hysteresis, Hofmeister series

## Abstract

Ion channels exhibit gating behavior, fluctuating between open and closed states, with the transmembrane voltage serving as one of the essential regulators of this process. Voltage gating is a fundamental functional aspect underlying the regulation of ion-selective, mostly α-helical, channels primarily found in excitable cell membranes. In contrast, there exists another group of larger, and less selective, β-barrel channels of a different origin, which are not directly associated with cell excitability. Remarkably, these channels can also undergo closing, or “gating”, induced by sufficiently strong electric fields. Once the field is removed, the channels reopen, preserving a memory of the gating process. In this study, we explored the hypothesis that the voltage-induced closure of the β-barrel channels can be seen as a form of reversible protein denaturation by the high electric fields applied in model membranes experiments—typically exceeding twenty million volts per meter—rather than a manifestation of functional gating. Here, we focused on the bacterial outer membrane channel OmpF reconstituted into planar lipid bilayers and analyzed various characteristics of the closing-opening process that support this idea. Specifically, we considered the nearly symmetric response to voltages of both polarities, the presence of multiple closed states, the stabilization of the open conformation in channel clusters, the long-term gating memory, and the Hofmeister effects in closing kinetics. Furthermore, we contemplate the evolutionary aspect of the phenomenon, proposing that the field-induced denaturation of membrane proteins might have served as a starting point for their development into amazing molecular machines such as voltage-gated channels of nerve and muscle cells.

## 1. Introduction

In recent decades, significant efforts have been devoted to unraveling the molecular mechanisms behind membrane channel gating. The success stories in understanding the voltage gating of highly ion-selective channels, mostly α-helical, of excitable cell membranes [1,2,3,4] have inspired numerous investigations into the structural aspects of voltage-induced closing in a distinct class of channels: the larger, less ion-selective β-barrel channels. These channels encompass various outer membrane proteins found in Gram-negative bacteria, mitochondria, and chloroplasts, as well as exotoxins secreted by bacteria targeting mammalian cells. While the β-barrel channels are not typically associated with the excitability phenomenon, experiments involving their reconstitution into planar lipid bilayers have shown that they can also be closed by applying sufficiently high voltages to the lipid bilayer. Despite the voltage “gating” being demonstrated in β-barrel channels of diverse origins and biological functions more than four decades ago, the underlying mechanisms and potential physiological significance of this phenomenon remain unresolved [5,6,7,8,9,10,11,12,13]. Furthermore, it remains unclear how much commonality exists between the voltage gating observed in the α-helical ion channels of excitable cells and that in the larger β-barrel channels. Equally uncertain is the extent of similarity in voltage gating among β-barrel channels of different origins. Notably, the physiological significance of β-barrel voltage gating, including both general and specific diffusion porins of Gram-negative bacteria and bacterial exotoxins, has been a subject of extensive debate for several decades, with Hiroshi Nikaido questioning it as early as 1988 by referring to the phenomenon as “an artifact of in vitro reconstitution” [14].

In the present study, we reexamine the fundamental aspects of voltage-induced closing in β-barrel channels, focusing on the outer membrane protein F (OmpF), which forms trimeric channels [15] in the outer membrane of the Gram-negative bacterium *Escherichia coli* (*E. coli*) [16,17]. Gram-negative bacteria are surrounded by a double-membrane envelope, where the outer membrane acts as a selective filter, providing protection against harmful components. Within this membrane, various types of β-barrel channels are found, with the OmpF trimer serving as a classic example. The OmpF trimeric structure has been determined with a resolution of 0.24 nm [18]. This protein can be isolated and reconstituted into planar lipid membranes (PLMs), allowing for examination at the single-channel level [6]. Based on our analysis of voltage-induced closing of the OmpF and data on other porins reported by our laboratory [19] and others previously [6,20,21,22,23,24], we propose that the “voltage gating of β-barrel channels” might represent their reversible denaturation induced by the applied electric field [19]. For a typical 100 mV transmembrane potential difference, this electric field reaches approximately 2 × 10^7^ to 3 × 10^7^ volts per meter. What is the magnitude of the forces acting on the channel-forming molecules in these fields? First, let us recall that a membrane with membrane-embedded proteins represents a capacitor, experiencing a compressive force between its “plates”. A rough estimate indicates that at 100 mV, this force, acting on an area of 10 nm^2^ (which is close, within an order of magnitude, to the solution-exposed surface of a typical β-barrel protein), is a fraction of one piconewton (pN). Importantly, this force is strong enough to be detectable in experiments on membrane voltage-induced thinning [25,26]. However, it is much smaller than the force of the field acting on any single charged residue within the channel-forming molecule. A quick estimate suggests that at the field strengths mentioned above, this force is an order of magnitude stronger than the compressive force, being in the range of 3 to 5 pN. Considering that at neutral pH, the OmpF molecule contains dozens of charged residues [27,28,29], this estimate indicates that a transmembrane potential difference of 100 mV exerts significant stress on the channel structure.

Here, we analyze several key observations, both made in this study and reported previously, that point towards a denaturation mechanism of voltage-induced closing of OmpF. These observations include the channel’s almost symmetric response to voltages of opposite polarity, the presence of multiple closed conformations, evident in the broad residual conductance distribution, the enduring memory of closing, the stabilization of open conformations in channel clusters, and the Hofmeister effect. We believe that the denaturation hypothesis provides an explanation for the slow progress in understanding the molecular mechanisms of β-barrel channel response to voltage. In many cases, voltage gating of ion-selective channels in excitable cells has been successfully linked to well-defined structural changes between the open and closed conformations of the channel-forming protein molecule. Conversely, it is widely recognized that protein denaturation, as well as the folding into functional structures, is a very complex process. This process evolves along various trajectories within a multidimensional energy landscape, often characterized by a plethora of poorly defined unfolded states, with the possible exception of small proteins [30,31].

## 2. Results

Figure 1A illustrates the voltage response of a single trimeric OmpF channel reconstituted into a PLM. At 75 mV, the trimer exhibits a rather stable current, regardless of the polarity of the applied voltage, over a timescale of several minutes. When subjected to 150 mV of either polarity, the channel undergoes closing in discrete steps, each reducing its initial conductance by approximately one-third. This behavior is considered archetypal and has been extensively documented and reviewed in previous studies [6,8,9,13,32,33,34]. Two notable features should be highlighted: first, the closing requires a relatively large transmembrane voltage difference; second, the channel responds to both negative and positive voltages. Figure 1B demonstrates the response of a multichannel OmpF system to the application of a slowly varying piecewise-linear periodic voltage ramp with an amplitude of ±200 mV.

The response is non-Ohmic, as higher voltages lead to the closing of some channels. In Figure 1C, we present the raw ion current data obtained from the same system at different frequencies of the ramp. Notably, at small voltages, all curves overlap, indicating that the number of channels in the membrane remains relatively constant throughout the experiment. At higher voltages, the curves diverge, exhibiting frequency-dependent hysteresis. The area of hysteresis increases as the ramp frequencies decrease from 5 to 1 mHz, meaning that at lower ramp frequencies, more channels have sufficient time to respond to the voltage. In certain parts of the records, the system displays negative differential resistance when an increase in the applied voltage results in a decrease in the current. Importantly, even in the case of the slowest ramp with a period of 1000 s (frequency of 1 mHz), the branches of the current records corresponding to increasing and decreasing voltages do not overlap. This observation suggests that the channels formed by OmpF exhibit long conformational memory, with characteristic times comparable to or significantly exceeding the 1000 s period.

In the study of the voltage-dependent anion channel (VDAC) from the outer mitochondrial membrane, another classic example of a β-barrel channel [10], a widely adopted method for quantifying the voltage-induced gating was pioneered by Marco Colombini over three decades ago [35]. This method is based on the analysis of VDAC’s response to slowly changing voltage ramps, similar to the experiments with the OmpF illustrated in Figure 2. Colombini’s approach focuses on the “opening” branches of the current-voltage relationship rather than the “closing” ones, as it was observed that VDAC opening generally occurs much faster than its closing. More recent studies involving the analysis of multichannel VDAC conductance kinetics in response to voltage jumps of approximately 30 mV amplitude have indeed confirmed this trend [36]. It was found that at voltage jumps of this amplitude, the opening process is approximately two orders of magnitude faster than the closing process. Specifically, the characteristic times for opening and closing were determined to be 0.8 s and 60 s, respectively.

As part of the present investigation, we applied this strategy to analyze the OmpF channels, with the results shown in Figure 2. We applied positive-polarity voltage ramps with an amplitude of 150 mV, covering a range of frequencies from 0.25 mHz to 10 mHz. Figure 2A provides the raw data for ion current through a multichannel system at a ramp frequency of 1 mHz. While the voltage gating effect is visibly lower than that observed with the 200 mV ramp in Figure 1, it remains measurable, as illustrated in Figure 2B and quantified in Figure 2C, using the hysteresis area as a metric. In line with the approach used in VDAC studies, we normalized the raw data with the following transformation [35,37].
(1)$$G_{norm}\left(V\right)=\left[G\left(V\right)-G_{min}\right]/\left[G_{0}-G_{min}\right],$$
where $G\left(V\right)$ is the multichannel conductance measured at voltage V, $G_{0}$ is the multichannel conductance at ~0 mV where most of the channels are open, and $G_{min}$ is the minimal multichannel conductance along the hysteretic cycle. To our surprise, in contrast to the results obtained for VDAC [36], not only did the opening branches coincide, but the closing branches of the hysteresis curves measured at different ramp frequencies showed nearly complete overlap. This is demonstrated in Figure 2D, suggesting a form of “quasi-equilibrium-like” behavior wherein the rate of voltage change remains slow enough for all ramp frequencies in the range of 0.25–5 mHz.

By fitting the data to the Boltzmann distribution
(2)$$G_{norm}\left(V\right)={\left\{1+exp\left[ne\left(\left|V\right|-V_{0}\right)/\left(k_{B}T\right)\right]\right\}}^{-1},$$
where $k_{B}$, T, and e represent the Boltzmann constant, absolute temperature, and elementary charge, respectively, we determined parameters for the effective “gating charge” (n) and the voltage of half-effect ($V_{0}$).

These parameters, as shown in Figure 2E,F, indicate the virtual independence of the ramp frequency. However, while the effective gating charges for both opening and closing processes exhibit only slight differences and are both found to be close to 1, the voltages of half-effect differ by a factor of 3. Specifically, the voltage of the half-effect for the opening branches is approximately 50 mV, whereas for the closing branches it is around 150 mV. This notable discrepancy indicates that the system possesses a long-time memory with a characteristic time significantly exceeding the largest period of the ramps used for voltage response measurements, which, in our case, is 4000 s. This observation aligns with previous reports of long relaxation times of this magnitude in other β-barrel bacterial porins [38]. It thus clearly demonstrates that the system is far from true equilibrium and, under the described experimental conditions, explores a subset of metastable states.

A similar conclusion can be drawn from the conductance relaxation experiments depicted in Figure 3. Both the processes of partial channel closing upon applying a 120 mV voltage and partial channel reopening upon returning to this voltage after holding the channels at 180 mV exhibit relatively fast relaxation components with characteristic times of about 40 and 5 s, respectively. An even faster component of the opening is seen at 100 s, upon reducing the voltage to 120 mV. However, following these fast components, there are long-term relaxations with characteristic times of many hundreds or thousands of seconds. Notably, this observation stands in stark contrast to the results published for VDAC voltage gating [36]. In Figure 3b of Ref. [36], it is shown that despite being two orders of magnitude different in their characteristic times, both closing and opening time dependences tend to converge to the same conductance.

The extended memory observed in the OmpF channels, spanning hundreds or even thousands of seconds, sets their voltage sensitivity apart from that of channels of excitable membranes. Indeed, complex kinetic behavior has also been observed for the latter and has been the subject of intense discussion for over four decades, dating back to the seminal study by Francisco Bezanilla and colleagues [39]. Over time, the functional significance of this complexity in the context of excitability phenomena has been well acknowledged [4,40]. This complexity sparked extensive theoretical efforts, with some approaches involving multi-state discrete models [40,41,42,43]. Others explore diffusion across a large number of states [44] or continuous diffusion within complex energy landscapes [45,46,47,48]. However, it is worth noting that the recovery times observed in experiments with bona fide voltage-gated channels [39,41,49] are typically much shorter than those we report here for OmpF.

The application of sufficiently high voltages to single OmpF channels induces their closure in three non-uniform increments [15] and, notably, results in a broad distribution of the “closed” state conductance [24]. This is illustrated in Figure 4, which provides statistics of the closure amplitudes. To facilitate comparison, all current histograms in panels B, C, D, E, and F are displayed within the same current interval of 250 pA. It is evident that not only the three increments corresponding to closings of individual pores within the trimer but also the “completely” closed state (Figure 4F) exhibit distributions of residual current that are significantly wider than those of the fully open state (Figure 4B). Notably, if normalized by the mean current, the relative width of the distribution in the closed state surpasses that of the open state by two orders of magnitude. This observation implies a high degree of disorder in the protein conformations representing the closed state.

We observed that the rate of the voltage-induced transition to the closed state is contingent on the ion species and follows the Hofmeister series [50,51] (Figure 5). In our experimental approach, we applied a 150 mV potential to a single OmpF channel and measured the time spent in its fully open conformation [19]. This procedure was repeated multiple times to collect the necessary statistics. Figure 5A provides three examples of such repetitions, while the corresponding histograms of the channel survival time in the fully open state are presented in Figure 5B for 1 M LiCl (*left*), KCl (*middle*), and RbCl (*right*) electrolyte solutions. Figure 5C summarizes the results obtained for several different salts, effectively demonstrating the Hofmeister effect in the rates of voltage-induced channel closing. The effect is also readily discernible in the hysteresis data plotted in Figure 5D,E for cations, and in Figure 5F,G for anions. Figure 5 shows both raw data (Figure 5D,F) and the areas encircled by the hysteresis curves (Figure 5E,G). Historically, the Hofmeister effect has been associated with protein stability and denaturation phenomena [50,51,52]. While the effect has been demonstrated in the context of channel-facilitated transport, to the best of our knowledge, it has not previously been linked to changes in channel conformations [53,54,55]. Although it was previously observed that not only the bathing electrolyte composition [19,56,57,58] but also the electrolyte concentration [19,59,60] can influence β-barrel channel gating, in the present study, we limited ourselves to 1 M salt solutions because our goal here was to demonstrate the existence of the Hofmeister effect in the voltage-induced OmpF closure.

It is well-known that the voltage-induced closing of β-barrel channels depends on the composition of lipid membranes [60,61,62,63,64,65,66,67,68]. In our multichannel experiments, we have observed that the voltage sensitivity of β-barrel channels, as illustrated here with the OmpF, also depends on the manner in which the channels are organized within the membrane. Specifically, our findings reveal that when channel insertion occurs in clusters, the transition into closed conformations is impeded compared to single channels or systems with multiple non-clustered channels. This is something to be expected because of the strong protein-protein interactions revealed in the early electron microscopy studies of OmpF by Jurg Rosenbusch and colleagues [69]. Indeed, if the voltage-induced collapse of the OmpF structure involves significant changes in the geometry of the outer protein surface, such collapse could be hindered by the tight packing of channels into clusters. A possible role of channel clusterization in the β-barrel channel gating was hinted at in the studies on VDAC regulation by a pro-apoptotic protein, Bid [70], and by non-lamellar membrane lipids [61]. Figure 6 demonstrates that when OmpF insertions occur in blocks comprising two or more channels (Figure 6C,E), the closing process does not proceed to the nearly zero current state (Figure 6A) typically observed in single-channel measurements; such behavior is illustrated in Figure 4A and Figure 5A.

As depicted in Figure 7, most of our findings can be tentatively rationalized by illustrating a one-dimensional cross-section of a highly multidimensional energy landscape of the proposed voltage-induced denaturation of the channel structure. At zero applied transmembrane voltage (*red curve*), the molecule’s lowest energy state corresponds to the folded state (F) representing the fully open channel conformation. When a large external field is imposed, for example, by the application of 150 mV of transmembrane voltage, the energy profile shifts (*blue curve*), favoring a family of (partially) unfolded states (U). Each unfolded state U exhibits significantly reduced conductance. These multiple unfolded states are denoted by the symbols i, i + 1, i + 2…

It is important to note that the energy landscape undergoes a similar transformation when voltages of either polarity are applied (as exemplified by Figure 1), resulting in energetically favorable channel states that fall within the unfolded category. The figure illustrates only one of the many possible cross-sections of the landscape. In a different cross-section, the state i + 1 could be the most readily accessible state for a direct transition from the folded state F. This limitation is inherent in any attempt to represent the complex multidimensional energy landscape of protein folding in one or two dimensions [30,31]. These multiple unfolded states are metastable, separated by barriers that are typically high enough for these states to be observed within the characteristic times of experiments. However, these barriers are not excessively high, allowing for spontaneous transitions between states with different conductance levels within the array of “closed” states U. An illustrative example of such transitions is shown in Figure 4A and Figure 5A. According to our model, these transitions may also underlie the long memory effects demonstrated in Figure 2 and Figure 3, enabling the channel to explore the vast space of closed states. At times, an OmpF monomer may transition to an exceptionally stable low-energy state in U. This phenomenon accounts for situations in which certain channels remain closed for several hours, even after the denaturing electric field has been removed (Appendix A). Paradoxically, under these circumstances, the subsequent application of high voltages of the opposite polarity, known to induce channel closure, may aid in channel reopening [19]. If our hypothesis of reversible denaturation is correct, further investigation is needed to understand how this stress promotes the refolding of the channel to its native conformation.

Overall, the OmpF memory characteristics were found to be even more complex. Over a period of extended time, the sensitivity of the OmpF closing to voltage exhibited a gradual increase in its magnitude (Appendix A). It appeared that the channels undergo a form of “training” when subjected to either repeated 0–150 mV voltage ramps (Appendix A) or a constant 150 mV (Appendix A), resulting in an enhancement of the voltage-induced closing effect as a function of time. To investigate whether the observed enhancement results from membrane or channel aging over an extended period, we conducted experiments in which the membrane was held at 0 mV of applied voltage for many hours between three repeated 0–150 mV voltage ramp applications (Appendix A). Appendix A provides a comparison of the results obtained from these three protocols. Notably, while the hysteresis area increased by nearly 30% (*left*) and 40% (*middle*) after 6 h of ramp application (Appendix A) and the voltage clamp protocol (Appendix A), respectively, a modest 14% increase (*right*) in the hysteresis area was observed during the third protocol, where the membrane was held at 0 mV applied voltage for a significantly longer time interval (15 h). The interpretation of the “channel training” effect within the context of the reversible protein denaturation model is not immediately clear.

The applied voltage reduces the height of the barrier for the channel to undergo unfolding (Figure 7). According to the data in Figure 5, the barrier height is also influenced by the cation and anion species following their position in the Hofmeister series. However, whether such dependence extends to the structural characteristics of the unfolded states’ energy landscape remains an open question. The long relaxation times observed in Figure 2 and Figure 3 present a challenge, as they hinder reliable quantification of the true equilibrium between the folded (F) and unfolded (U) states. Remarkably, the Hofmeister effect in OmpF voltage-induced closing also exhibits an intriguing memory effect, both at the single-channel and multichannel levels (Appendix A). When CsCl electrolyte solutions were replaced by LiCl without compromising membrane integrity, the channel closing kinetics displayed a gradual change in the expected direction but remained quite fast, failing to reach the level observed in LiCl solutions when used from the beginning of the experiment, including the initial channel reconstitution (Appendix A). At the same time, following the solution exchange, the channel conductance aligned with that in LiCl. The same was true for the LiCl to CsCl replacement (Appendix A).

The results presented in Figure 6 strongly support the notion that the folded structure of the channel experiences stabilization when organized within channel clusters. This finding lends credence to the idea that the voltage-induced closing of the OmpF channel involves substantial structural changes, akin to what has been suggested for the voltage-induced closing of another β-barrel channel, VDAC [71]. The rationale behind this conjecture lies in the belief that if the voltage-induced closing were a result of relatively subtle changes within the barrel, such as, e.g., the motion of the L3 loop [72,73,74,75], screening/unscreening of charges within the channel lumen [23,76], or some “breakdown of the delicate ion-conducting pathway” [8], the presence of neighboring channels would not measurably interfere with the process.

## 3. Discussion

In Table 1, we present a compilation of empirical findings of voltage-induced closing in OmpF channels. These observations, both from our study and prior work, are divided into two categories: those supporting voltage-induced OmpF denaturation and those supporting functional gating. While most findings lean towards denaturation, we entrust the decision to the readers, allowing them to draw their own conclusion about the mechanism of large β-barrel channel voltage gating.

Our study’s driving motive was to find the “light at the end of the channel” by providing a plausible explanation for the stubborn mystery surrounding the voltage-induced closing of large β-barrel channels. In the present work, we explored the possibility that this phenomenon results from the denaturation of the channel-forming protein under the influence of applied electric fields, thus attempting to explain the enduring difficulties in identifying specific structures of, or even intermediates leading to, the closed states. This stands in contrast to functional voltage gating, which is a characteristic feature of ion-selective channels of excitable membranes—channels that play a crucial role in nerve pulse propagation and have been explored in an impressive bulk of celebrated research [1,2,3,4,77,78,79,80,81,82,83,84]. Our analysis, based on the data of the present and many previous studies, consistently leans towards denaturation, as argued throughout the text and summarized in Table 1. This concept was previously hinted at by Robertson and Tielemen [9] and explored in greater detail in our conference proceedings publication [19]. Within this framework, the applied fields of more than 2 × 10^7^ V/m, corresponding to a 100 mV electric potential difference across the lipid bilayer, crash the native protein structure into a set of partially unfolded, less conductive conformations. Once the denaturing field is removed, OmpF can regain its native function by refolding back to its original open-channel conformation.

After considering all the pro- and contra-arguments, a reasonable question to ask is whether there exists a well-defined boundary between the phenomenon of functional voltage gating and that of voltage-induced denaturation. Our tentative answer is that this boundary is probably quite vague. One of the sources of this ambiguity stems from the possible evolutionary connection between these two phenomena. It is conceivable that, as an ancient mechanism of voltage sensitivity, nature might have employed reversible denaturation of membrane proteins in response to the transmembrane electric field. Over time, in this scenario, evolutionary pressures would have transformed these denaturation-based voltage-sensitive “proto-channels” into the perfect voltage-sensing molecular machines that exhibit distinct structures in both their open and closed states [1,2,83].

Could voltage-induced closure of OmpF function as a crucial attribute in optimizing nutrient uptake and waste product release while also serving as a defense mechanism against the infiltration of foreign molecules, like antibiotics, by controlling the number of open channels? Considering that a single *E. coli* cell contains approximately 10^5^ copies of OmpF channels [85], alongside various other general and specific porin channels, it is evident that alteration in porin expression can influence outer membrane permeability [86]. However, could voltage-triggered protein denaturation achieve a similar outcome? Could a voltage of sufficient magnitude to induce channel closing even be attained across the outer membrane? For some *E. coli* outer membrane porins, such as OmpC [87], the “crucial voltage”, a vague term intentionally sidestepped in the present article, was reported to be close to 200 mV, whereas other porins, e.g., the recently discovered putative β-barrel channel Triplin [88] and *Vibrio cholerae* OmpT exhibit voltage-induced closing at considerably lower transmembrane voltages [89]. At the same time, an increase in solution acidity [90] and the presence of polyamines, such as spermine, spermidine, and cadaverine, are cofactors known to increase the voltage sensitivity of OmpF [16,22,91]. Finding answers to these questions is not straightforward and requires further investigation and critical thinking. We should also acknowledge recent progress in our understanding, or more accurately, our increasing recognition of the complexities of membrane potential dynamics [92]. Although the Donnan potential was initially believed to have no impact on porin permeability [14], it is now recognized as a parameter that can dynamically vary based on factors like osmotic strength, external pH, and changes in cell surface charges [93]. Additionally, electrical spiking in *E. coli*, which is responsive to various chemical and physical influences and aligns with the rapid efflux of small molecules, has also been documented [94]. Drawing a comparison between bacteria and eukaryotes, the authors emphasized the high surface-to-volume ratio characteristic of bacteria and proposed to reconsider certain principles in neuronal electrophysiology within the bacterial context. For instance, in a bacterium with a volume of 1 fL and a cytoplasmic Na^+^ concentration of 10 mM, there are only approximately ~10^7^ ions of Na^+^ available. This implies that a single ion channel, carrying a current of 2 pA, could deplete this ion supply in less than 1 s. It was suggested that by oscillating back and forth, the bacterium might be dynamically changing its membrane potential to various values, each of which is optimal for a distinct biological process.

Finally, we hypothesize that the major channel of the outer mitochondrial membrane, VDAC [10], may have evolved to incorporate its voltage-induced denaturation as at least a partially functional feature. However, the question of whether mitochondrial outer membrane voltages alone are sufficient to gate VDAC independently of its cytosolic regulators [95,96,97] remains a topic of intense debate (see discussions in Refs. [98,99,100]).

There are additional reports showing that β-barrel channels can undergo gating under the application of low, physiologically relevant voltages [88,89], thus suggesting gating functionality, or exhibiting a prominent asymmetry toward the applied voltage sign [72,101,102,103,104]. Over the past 25 years, researchers from our laboratories have collectively investigated eleven distinct β-barrel channel-forming proteins, encompassing five outer membrane bacterial porins, five bacterial exotoxins, and mitochondrial VDAC. All of these proteins exhibited reversible voltage-dependent closings. While certain distinctions were noted, such as in the case of the anthrax toxin which operates amidst significant endosomal voltage and pH gradients and is characterized by a pronounced asymmetry in response to the polarity of the applied voltage [101], many aspects of this process were found to be strikingly similar to other β-barrel channels.

## 4. Materials and Methods

### 4.1. Chemicals

Wild-type OmpF were generously provided by Dr. Mathias Winterhalter (Jacobs University, Bremen, Germany). The protein was diluted to a concentration of 0.22, 2.2, and 22 µg/mL using 1 M KCl and 1% (*v*/*v*) n-octylpolyoxyethylene (octyl-POE) solution (Alexis Biochemicals, Lausen, Switzerland). The following chemical reagents were used: LiCl, NaCl, KCl, RbCl, CsCl, 4-(2-hydroxyethyl)piperazine-1-ethanesulfonic acid (HEPES), LiOH, NaOH, KOH, RbOH, CsOH (Sigma-Aldrich, St. Louis, MO, USA), “purum” hexadecane (Fluka, Buchs, Switzerland), diphytanoylphosphatidylcholine (DPhPC) in chloroform (Avanti Polar Lipids, Alabaster, AL, USA), pentane (Burdick and Jackson, Muskegon, MI, USA), and agarose (Bethesda Research Laboratory, Gaithersburg, MD, USA). The 1 M LiCl, NaCl, KCl, CsCl, and RbCl solutions were buffered at pH 7.4 by 5 mM HEPES. The pH of stock solutions was individually adjusted by adding LiOH, NaOH, KOH, RbOH, or CsOH as needed. All solutions were prepared using double-distilled water.

### 4.2. Channel Reconstitution

“Solvent-free” planar lipid bilayer membranes were formed following the lipid monolayer opposition technique [105] from a 5 mg/mL solution of DPhPC in n-pentane on a 60 µm diameter aperture in the 15 µm thick Teflon film that separated two (*cis* and *trans*) compartments of the custom-made Teflon chamber. Electrolyte solutions were symmetrically added to both sides of the chamber, and measurements were performed at T = 23 ± 0.5 °C. The aperture was pretreated with a 1% solution of hexadecane in pentane and dried for 15 min prior to membrane formation. The film and the total capacitances were close to 25 and 50 pF, respectively. Single channels were reconstituted by adding 0.1–0.3 µL of 0.22 µg/mL stock solution of OmpF to the 1.5-mL aqueous phase in the *cis* half of the chamber while stirring at 150 mV of applied voltage for 5–10 min. In the multichannel bilayer experiments, channels were typically formed by adding 2–3 µL of a 0.22 µg/mL stock solution of OmpF prepared several weeks in advance to the *cis* compartment while stirring. To investigate the effect of OmpF clustering on voltage-induced closing (Figure 6), we compared freshly prepared with properly stored at +4 °C eight-week-old 0.22 µg/mL OmpF solutions and used more concentrated 22 µg/mL OmpF. Steady multichannel conductance, monitored by applying a 100 mV transmembrane voltage, was achieved in about 30 min. The electric potential difference across the membrane was applied with a pair of Ag-AgCl electrodes in 2 M KCl and 1.5% agarose bridges. The potential was considered positive when it was greater on the side of OmpF addition (*cis* side). Current recordings were obtained using an Axopatch 200B amplifier (Molecular Devices, San Jose, CA, USA) in the voltage-clamp mode. All experiments were carried out at a room temperature of 23.0 ± 1.5 °C. Single-channel data were filtered using a low-pass eight-pole Butterworth filter (Model 900 Frequency Active Filter, Frequency Devices, Ottawa, IL, USA) at 15 kHz, acquired with a Digidata 1322A board (Molecular Devices, San Jose, CA, USA) at a sampling frequency of 50 kHz, and analyzed using pClamp 10.7 software (Molecular Devices, San Jose, CA, USA). Multichannel data were saved with a sampling frequency of 2–5 kHz; the low-pass Bessel filter was set to 1 kHz.

### 4.3. Voltage Gating Measurements

Five different experimental protocols were used to measure OmpF voltage-induced closing. *First*, hysteresis experiments [36] were conducted using periodic triangular waves applied from a Function Waveform Generator 33120A (Hewlett Packard, Palo Alto, CA, USA) with voltage changing from 200 mV to −200 mV (Figure 1B) or from 0 to 150 mV (Figure 2A) and back in the frequency range of 0.25–10 mHz. The current recordings were collected from the membranes containing 30–200 channels and averaged over several periodic triangular waves applied to the same membranes. *Second*, OmpF multichannel closing and opening kinetics were investigated using the previously described conductance relaxation experiments (see Figure 3 in Ref. [36]), with some modifications. The details of the “conductance relaxation to the same voltage” protocols are given in Figure 3 legends. *Third*, to quantify the distribution of the voltage gating event currents, we recorded the closings of all three monomers of a single OmpF trimer applying 150 mV voltage (Figure 4). *Fourth*, the stability of *single* OmpF channels in response to 150 mV transmembrane voltage was studied by measuring the time needed for the closing of one (first) monomer in the trimer (Figure 5). In Figure 4 and Figure 5, after each measurement, 0 mV (or occasionally −150 mV) was applied to reopen the channel; the measurements were repeated multiple times and quantified by fitting the closing time histograms with a single exponent. *Fifth*, we conducted comparative multichannel PLM measurements by reconstituting OmpF from freshly diluted and eight-week-old solutions at concentrations of 0.22 and 22 µg/mL. We initially reconstituted the channels under an applied voltage of 50 mV and subsequently examined the voltage-induced closing effect at 150 mV (Figure 6).

## Figures and Tables

**Figure 1 ijms-24-16655-f001:**
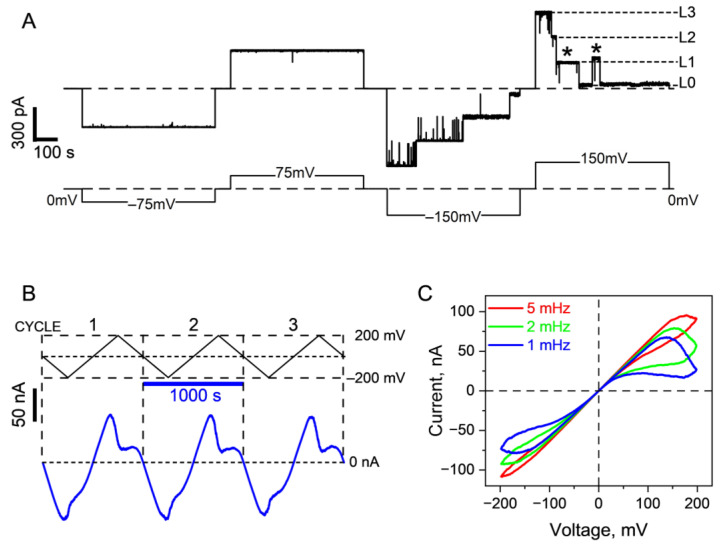
Voltage-dependent closing-opening transitions of OmpF were observed at the single-channel (**A**) and multichannel (**B**,**C**) levels. (**A**) Typical recordings of ion current through a single trimeric OmpF channel reconstituted into PLMs taken at ±75 mV and ±150 mV applied voltage. Notably, higher voltages lead to the channel closing in approximately three equal but not identical steps. Dashed lines indicate zero current (*top*) and voltage (*bottom*) levels. Short-dashed lines designate fully open (L3), one monomer closed (L2), two monomers closed (L1), and residual current (L0) states, with two asterisks (*) highlighting current variations in the L1 state. (**B**) Raw ion current data from a multichannel system containing around 75 OmpF channels (*bottom*), subjected to a three-fold repetition of a −200 mV to +200 mV voltage ramp at 1 mHz frequency (*top*). (**C**) I-V curves, replotted from the experiments presented in panel (**B**), at three frequencies (5 mHz, 2 mHz, and 1 mHz), averaged over three ramp cycles, elucidating voltage gating at both polarities. The ion channel recordings were additionally filtered using a 1000 Hz (**A**) and 100 Hz low-pass Bessel filter (**B**,**C**), and data reduction (reduction factor 100, Clampfit, Molecular Devices, San Jose, CA, USA) was applied. Here, the membrane bathing solutions contained 1 M KCl and 5 mM HEPES at pH 7.4.

**Figure 2 ijms-24-16655-f002:**
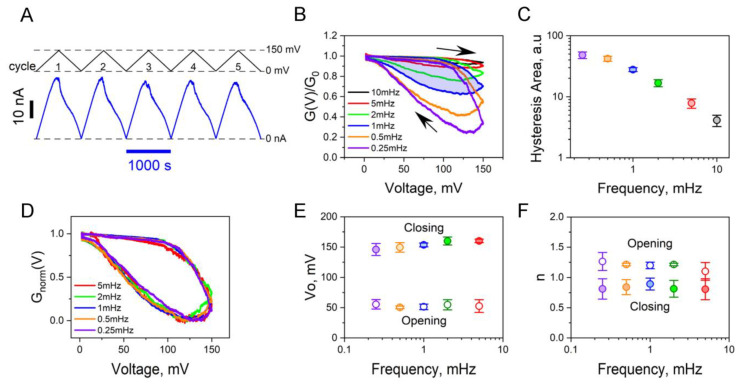
The conductance of a multichannel membrane, responding to a voltage ramp from 0.25 mHz to 10 mHz, reveals frequency-dependent sensitivity in its hysteresis curves. (**A**) Raw data showing ion current across approximately 75 OmpF channels (*bottom*) in response to a series of repeated 0 to 150 mV voltage ramps at a frequency of 1 mHz (*top*). (**B**) Conductance hysteresis curves at different ramp frequencies. Arrows indicate voltage change directions during channel closing and opening within the hysteresis loop. The data represents an average across 3–6 ramp periods. (**C**) Areas encircled by hysteresis curves from panel (**B**) as a function of ramp frequency. The data averaged over 3–5 independent experiments with 60–120 OmpF channels. (**D**) Conductance hysteresis curves from panel (**B**) after transformation according to Equation (1) in the text. Normalized hysteresis curves show that both the closing and opening branches of the hysteresis curves nearly overlap. (**E**,**F**) Gating parameters, the voltage of half-effect $V_{0}$ (**E**), and the effective “gating charge” n (**F**) obtained by fitting the opening and closing branches of the panel (**D**) curves to Equation (2) in the text, showing hardly any dependence on the ramp frequency.

**Figure 3 ijms-24-16655-f003:**
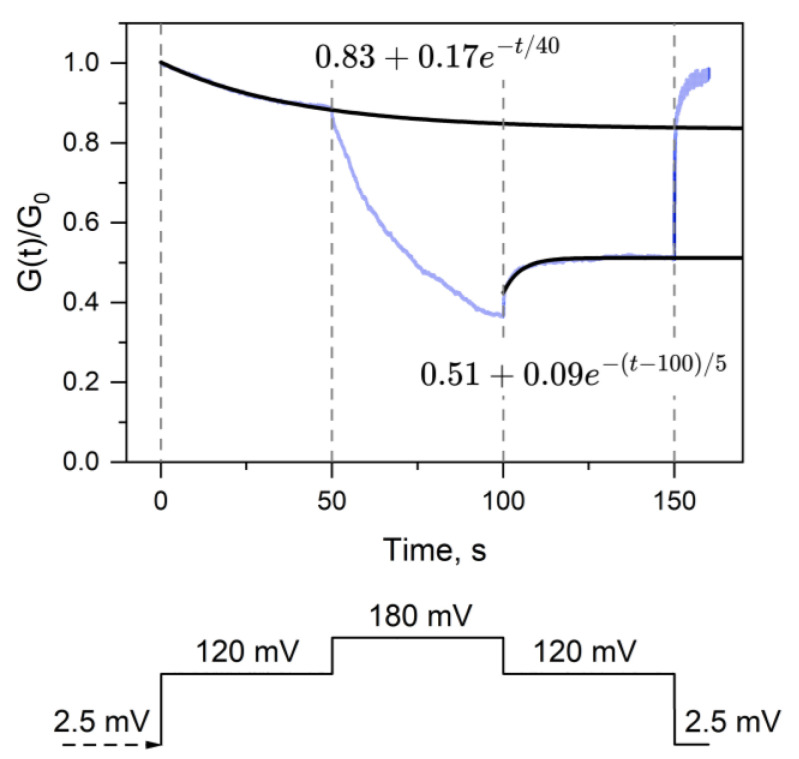
Multichannel conductance relaxation experiment. The experiment was started at the applied voltage of 2.5 mV, then, for durations of 50 s, it was switched to 120 mV, 180 mV, back to 120 mV, and then returned to 2.5 mV as shown in the voltage protocol at the bottom part. Solid lines depict the best-fit single exponentials in the 50 s intervals (the fast component of the opening branch at 100 s is ignored), with the equations displayed in the figure. The lines do not converge to the same value, illustrating the long-term memory of the OmpF voltage-induced closing and opening. The data represent averages from 10 runs. Recordings were filtered using a 100 Hz low-pass Bessel filter, and data reduction (reduction factor 100, Clampfit, Molecular Devices) was applied.

**Figure 4 ijms-24-16655-f004:**
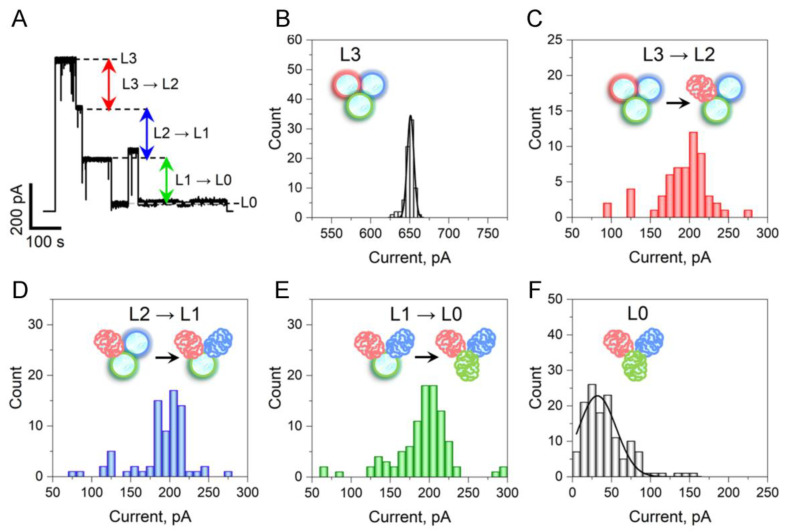
The current distribution of voltage-induced closing events reveals the variety of conformational states within the closed states of OmpF. (**A**) An experimental procedure for quantifying the current of the open OmpF ion channel (**B**), voltage-induced closing event currents (**C**–**E**), and residual conductance current (**F**). To mitigate current variations between channels, data were collected from the same single OmpF channel by repeatedly applying a 150 mV voltage and recording its 3-step closings, similar to those depicted in panel (**A**), until sufficient statistical data were accumulated. The current histogram from a single open OmpF channel (L3) shown in panel (**B**), acquired after closings under 150 mV and the subsequent reopenings at 0 mV, displays a narrow current distribution of 651 ± 9 pA. The current histograms in panels C-E of the 1st (L3 → L2), 2nd (L2 → L1), and 3rd (L1 → L0) monomer closings, with the arrows showing transitions, as well as of the residual current (L0) in panel (**F**), exhibit significantly broader ranges compared to the narrow distribution of the open state current. To ensure easy comparison, panels (**B**–**F**) are graphed with the equivalent 250-pA range on the *X*-axis. Interestingly, the closing event current histograms in panels (**C**–**E**) extend beyond one-third of the total channel current, suggesting that the consequences of a single monomer closing could impact nearby monomer conformations and their associated currents. Note that in panel (**F**), the single Gaussian fit is drawn to guide the eye only as it significantly extends into the unrealistic negative current values.

**Figure 5 ijms-24-16655-f005:**
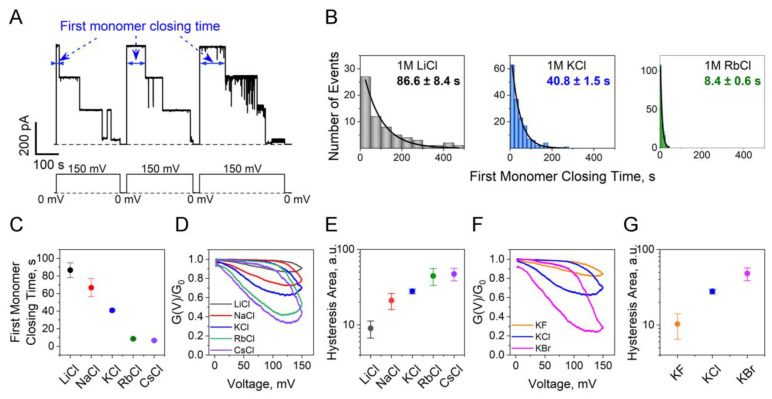
Hofmeister effect in OmpF voltage-induced closing. (**A**) Examples of raw currents through a single OmpF channel at the application of 150 mV. (**B**) The histograms of the first monomer closing times are fitted by a single exponential function. For illustrative purposes, only the 1 M LiCl (*left panel*), KCl (*middle panel*), and RbCl (*right panel*) data are shown. (**C**) There is an order of magnitude difference in closing times in the LiCl, NaCl, KCl, RbCl, and CsCl series. (**D**,**E**) Multichannel conductance hysteresis curves at 1 mHz ramp frequency (**D**) show an increase of more than four times in the areas encircled by hysteresis curves (**E**) in the LiCl, NaCl, KCl, RbCl, CsCl series. (**F**,**G**) Multichannel conductance 1 mHz hysteresis curves for different anions. The influence of anions on the OmpF voltage-induced closing and opening (**F**) is seen as a more than fourfold increase in the areas encircled by hysteresis curves (**G**) in the KF, KCl, and KBr series.

**Figure 6 ijms-24-16655-f006:**
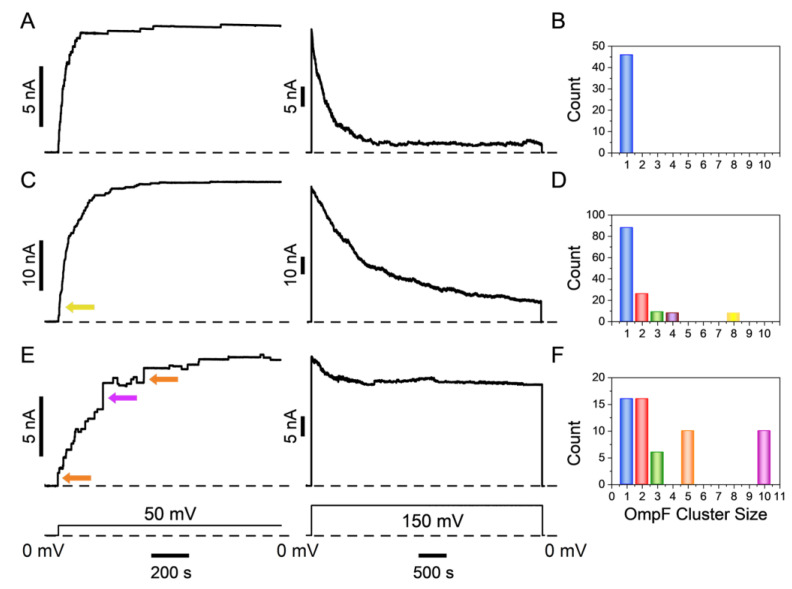
Impact of protein cluster insertion on voltage-induced closing. (**A**,**B**) A typical multichannel experiment wherein OmpF is exclusively inserted into the PLM as trimeric units, resulting in an average current of 0.23 ± 0.01 nA at 50 mV applied voltage. Notably, upon the application of 150 mV voltage, the channels in this multichannel membrane display a relatively swift and nearly complete closing. OmpF was added into the *cis* chamber of the bilayer chamber using a 0.22 μg/mL solution following a 1:10,000 stock solution dilution performed approximately eight weeks prior to the experiment. Panel (**B**) displays a cluster size distribution graph exclusively depicting trimeric OmpF single channel insertions. (**C**,**D**) A typical experiment in which about 35% of the total multichannel current was a result of the insertion of OmpF clusters. These clusters comprised thirteen dimers, three trimers, four tetramers, and one octamer, as seen from the cluster distribution graph in panel (**D**). The remaining current was generated by the insertion of 88 single-channel OmpF trimers. Notably, a significant reduction in the degree of channel closing is observed at 150 mV ((**C**), *right panel*). OmpF was added to the *cis* size of the bilayer chamber using a freshly diluted 0.22 μg/mL solution. (**E**,**F**) A typical experiment wherein the insertion of OmpF clusters accounted for over 70% of the total multichannel membrane conductance. These clusters consisted of eight dimers, three trimers, two pentamers, and one decamer (**F**). Upon applying a voltage of 150 mV ((**E**), *right panel*), the effect of voltage on channel closing noticeably diminishes, resulting in a substantial residual conductance that corresponds to approximately 74% of the initial current. OmpF was added to the *cis* size of the bilayer chamber using a freshly diluted 22 µg/mL solution. The lower panel illustrates the voltage protocol employed for panels (**A**,**C**,**E**). For illustrative purposes, in panel (**C**), a yellow arrow marks one octamer cluster insertion. In panel (**E**), orange arrows indicate two pentamer cluster insertions, while a magenta arrow marks one decamer cluster insertion. The same color code is used in panels (**B**,**D**,**F**). The ion channel recordings were filtered using a 50 Hz Bessel filter and data reduction (reduction factor 100, Clampfit, Molecular Devices) was applied.

**Figure 7 ijms-24-16655-f007:**
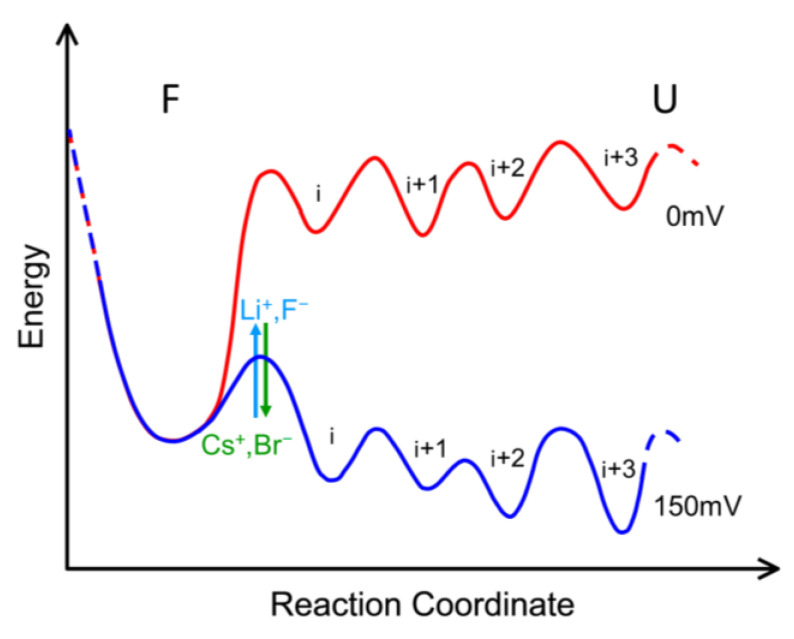
One-dimensional cross-section of an energy landscape illustrating voltage-induced channel protein denaturation. Hofmeister series ions are able to change the height of the barrier separating the functional folded state F and the multitude of (partially) unfolded ones U.

**Table 1 ijms-24-16655-t001:** Empirical Findings with the OmpF channel: Field-Induced Denaturation vs. Functional Gating.

Empirical Findings	In Favor of Field-Induced Denaturation	In Favor of Functional Gating	Source
Closing by strong electric fields	Yes	Maybe	Refs. [6,20,21,23], present study, Figure 1 and Figure 2
Response to voltages of either polarity	Yes	Unlikely	Refs. [6,20], present study, Figure 1
A complex array of voltage gating memory effects	Yes	Maybe	Ref. [19], present study, Figure 2, Figure 3 and Appendix A
Multiplicity of closed state conformations	Yes	Unlikely	Refs. [19,24], present study, Figure 4
Effects of salts along the Hofmeister series	Yes	Maybe	Ref. [19], present study, Figure 5 and Appendix A
Open conformation is stabilized in clusters	Yes	Unlikely	Present study, Figure 6

## Data Availability

The data presented in this study are available upon request from the corresponding author.

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
