# Peer review of "Beta-Barrel Channel Response to High Electric Fields: Functional Gating or Reversible Denaturation?"

_ijms, 2023, doi:10.3390/ijms242316655_

Round 1

Reviewer 1 Report

Comments and Suggestions for Authors

 In this article by E. M. Nestorovich and S. M. Bezrukov, the authors provide a different perspective on the functionality of beta-barrel channels closing under the influence of large electric fields by considering reversible denaturation as a potential mechanism of gating. The authors employed traditional electrophysiology measurements on bacterial OmpF channels reconstituted into artificial lipid membranes. They observed a nearly symmetric response to voltages of positive and negative polarities, identified multiple closed states and stabilization of the open conformation in channel clusters, and a long-term gating memory. These results question the classical model of voltage gating and support the hypothesis of a denaturation-based mechanism. Additional support for the hypothesis of denaturation is provided by experiments aimed at investigating the effects of ions on closing kinetics; interestingly, the ranked effects presented by the added ions follow the Hofmeister series.

Although the authors do not provide clear-cut evidence of denaturation (this is indicated early from the question mark in the title), I consider that this article provides a new perspective on the mechanisms by which certain channels may open and close in response to transmembrane electric fields. In my opinion, the article is suitable for publication in the International Journal of Molecular Sciences.

Nonetheless, I have some minor comments and suggestions for improving the clarity of the concepts, justifying the choice of experimental conditions, and prompting scientists to pursue follow up investigations intended to elucidate the mechanisms of gating of beta-barrel channels.   

  Comments/Suggestions:

-The authors may provide a better clarification of the distinction between functional gating and denaturation (lines 423-428). Functional voltage gating seems to be restricted to “ion-selective channels of excitable membranes”. Why such a limitation? Many other ion-channels in non-excitable membranes present voltage gating that occurs within physiological voltage range and their gating has clear physiological significance. I have no doubt that their gating mechanisms are similar to ion channels from excitable membranes (as opposed to denaturation). 

-The authors hypothesize that channel closing is a consequence of reversible denaturation induced by electric field as opposed to functional gating. However, they use the Boltzmann distribution to determine the “effective gating charge”, a feature common to functional gating. What would be the interpretation of this experimental parameter (i.e., gating charge) in the context of denaturation?

-In the same line, the I-V plots in Figure 1 show closing at both positive and negative voltages but the plots are not perfectly symmetrical; however, the next experiments reported in the article employed positive voltages. It is not clear if the parameters determined and shown in Figure 2 (and the following figures) would be the same in the negative voltage range (probably not); what the interpretation of the different values would be in the context of a gating mechanism originating in denaturation? This may be detailed in the discussion section.

-The authors consistently used a single electrolyte concentration (i.e., 1M) to demonstrate that the observed effects are consistent with the Hofmeister series. This raises the question of how salt concentration will affect the voltage-induced closing. A justification for the use of a single, high concentration may be provided.

-The gating of another beta-barrel channel with an obscure physiological role, lysenin, is not mentioned in the introduction section (i.e., its voltage gating and influence from added ions, and the long-term memory) and/or discussion section (i.e., in the context of refs. 82 and 83 - porins gating at low voltages, and ref. 95 - asymmetric response to voltages).

In conclusion, I consider that the article is suitable for publication in the International Journal of Molecular Sciences after a minor revision. 

Author Response

Response to Referee 1

In this article by E. M. Nestorovich and S. M. Bezrukov, the authors provide a different perspective on the functionality of beta-barrel channels closing under the influence of large electric fields by considering reversible denaturation as a potential mechanism of gating. The authors employed traditional electrophysiology measurements on bacterial OmpF channels reconstituted into artificial lipid membranes. They observed a nearly symmetric response to voltages of positive and negative polarities, identified multiple closed states and stabilization of the open conformation in channel clusters, and a long-term gating memory. These results question the classical model of voltage gating and support the hypothesis of a denaturation-based mechanism. Additional support for the hypothesis of denaturation is provided by experiments aimed at investigating the effects of ions on closing kinetics; interestingly, the ranked effects presented by the added ions follow the Hofmeister series.

Although the authors do not provide clear-cut evidence of denaturation (this is indicated early from the question mark in the title), I consider that this article provides a new perspective on the mechanisms by which certain channels may open and close in response to transmembrane electric fields. In my opinion, the article is suitable for publication in the International Journal of Molecular Sciences.

Thank you.

Nonetheless, I have some minor comments and suggestions for improving the clarity of the concepts, justifying the choice of experimental conditions, and prompting scientists to pursue follow up investigations intended to elucidate the mechanisms of gating of beta-barrel channels.  

  Comments/Suggestions:

-The authors may provide a better clarification of the distinction between functional gating and denaturation (lines 423-428). Functional voltage gating seems to be restricted to “ion-selective channels of excitable membranes”. Why such a limitation? Many other ion-channels in non-excitable membranes present voltage gating that occurs within physiological voltage range and their gating has clear physiological significance. I have no doubt that their gating mechanisms are similar to ion channels from excitable membranes (as opposed to denaturation).

Thank you for this comment. We do agree with the Referee. The absence of a clear-cut borderline between the functional gating and voltage-induced denaturation is commented on in Discussion, lines 455-464.

-The authors hypothesize that channel closing is a consequence of reversible denaturation induced by electric field as opposed to functional gating. However, they use the Boltzmann distribution to determine the “effective gating charge”, a feature common to functional gating. What would be the interpretation of this experimental parameter (i.e., gating charge) in the context of denaturation?

The Referee is right, strictly speaking the Boltzmann distribution is valid for equilibrium systems only. We are fully aware of this and discuss it in lines 192-201. However, fitting by the Boltzmann distribution can be justified when the parameters obtained from such fitting do not depend on the rate of change of the applied force. This happens to be the case in our experiments as is shown in Figs. 2E and 2F.

-In the same line, the I-V plots in Figure 1 show closing at both positive and negative voltages but the plots are not perfectly symmetrical; however, the next experiments reported in the article employed positive voltages. It is not clear if the parameters determined and shown in Figure 2 (and the following figures) would be the same in the negative voltage range (probably not); what the interpretation of the different values would be in the context of a gating mechanism originating in denaturation? This may be detailed in the discussion section.

We used only positive voltages just to illustrate/prove the idea that the system is far from equilibrium.  The following phrase is added in the revised version on page 5: “It thus clearly demonstrates that the system is far from true equilibrium and, under the described experimental conditions, explores a subset of metastable states”.

-The authors consistently used a single electrolyte concentration (i.e., 1M) to demonstrate that the observed effects are consistent with the Hofmeister series. This raises the question of how salt concentration will affect the voltage-induced closing. A justification for the use of a single, high concentration may be provided.

Thank you. The following sentence is added on page 8: “Although it was previously observed that not only the bathing electrolyte composition [19,56-58] but also electrolyte concentration [19,59,60] can influence β-barrel channel gating, in the present study we limited ourselves to 1 M salt solutions, because our goal here was to demonstrate the existence of the Hofmeister effect in the voltage-induced OmpF closure.”

-The gating of another beta-barrel channel with an obscure physiological role, lysenin, is not mentioned in the introduction section (i.e., its voltage gating and influence from added ions, and the long-term memory) and/or discussion section (i.e., in the context of refs. 82 and 83 - porins gating at low voltages, and ref. 95 - asymmetric response to voltages).

Thank you.  This is discussed now on page 13: There are additional reports showing that β-barrel channels can undergo gating under the application of low, physiologically relevant voltages [88,89], thus suggesting gating functionality, or exhibit a prominent asymmetry toward the applied voltage sign [72,101-104].

In conclusion, I consider that the article is suitable for publication in the International Journal of Molecular Sciences after a minor revision.

Thank you for appreciating our work.

Reviewer 2 Report

Comments and Suggestions for Authors

Nestorovich & Bezrukov studied the effect of high electric fields on the β-barrel channels, OmpF. The experiments are nicely done and the analysis seems very reasonable. The authors studied the the symmetric response of OmpF to voltages of both polarities, the stabilization of the open conformation of the channel, the long-term gating memory, and the Hofmeister effects in closing kinetics. They concluded that there is a multiple closed states of OmpF. They also proposed that application of high electric field induced denaturation of OmpF which may be correlated to long term gating memory.

Main concerns:

- As I mentioned earlier the experiments are well done and I have no comment on it. But my main concern is that what would be the benefit, for bacteria, of having a molecular machine that register memory in hundreds of seconds while the whole life span of a typical bacteria is 12 hours!

Author Response

RE:      “Beta-Barrel Channel Response to High Electric Fields: Functional Gating or Reversible Denaturation?” by E. M. Nestorovich and S. M. Bezrukov

Dear Editor,

We are grateful to you for handling our manuscript and to the Referees for reviewing our work.  All the points raised by the Referees are discussed in detail in our responses below. All changes made in the revised manuscript are highlighted in yellow.

We hope that the revised manuscript satisfies the criteria and will be accepted for publication in IJMS.

Thank you for your consideration,

Ekaterina Nestorovich and Sergey Bezrukov

Thank you for appreciating our work.

Response to Referee 2

Nestorovich & Bezrukov studied the effect of high electric fields on the β-barrel channels, OmpF. The experiments are nicely done and the analysis seems very reasonable. The authors studied the the symmetric response of OmpF to voltages of both polarities, the stabilization of the open conformation of the channel, the long-term gating memory, and the Hofmeister effects in closing kinetics. They concluded that there is a multiple closed states of OmpF. They also proposed that application of high electric field induced denaturation of OmpF which may be correlated to long term gating memory.

Main concerns:

- As I mentioned earlier the experiments are well done and I have no comment on it. But my main concern is that what would be the benefit, for bacteria, of having a molecular machine that register memory in hundreds of seconds while the whole life span of a typical bacteria is 12 hours!

Yes, we agree. This is one more strong argument in favor of “voltage gating” as a non-functional effect. However, we decided not to add it to the main text keeping in mind that in the dormant state, e.g., in biofilms, bacteria live “forever”, though this situation is not typical indeed.

Thank you.
